# Antimicrobial Resistance Patterns in Organic and Conventional Dairy Herds in Sweden

**DOI:** 10.3390/antibiotics9110834

**Published:** 2020-11-21

**Authors:** Karin Sjöström, Rachel A. Hickman, Viktoria Tepper, Gabriela Olmos Antillón, Josef D. Järhult, Ulf Emanuelson, Nils Fall, Susanna Sternberg Lewerin

**Affiliations:** 1Department of Clinical Sciences, Swedish University of Agricultural Sciences, 750 07 Uppsala, Sweden; karin.sjostrom@slu.se (K.S.); gabriela.olmos.antillon@slu.se (G.O.A.); ulf.emanuelson@slu.se (U.E.); nils.fall@slu.se (N.F.); 2Department of Medical Biochemistry and Microbiology, Zoonosis Science Center, Uppsala University, 751 23 Uppsala, Sweden; rachel.hickman@medsci.uu.se (R.A.H.); vtepper05@gmail.com (V.T.); 3Institute of Environmental Engineering, ETH, Stefano-Franscini-Platz 5, 8093 Zürich, Switzerland; 4Department of Medical Sciences, Zoonosis Science Center, Uppsala University Hospital, 751 85 Uppsala, Sweden; josef.jarhult@medsci.uu.se; 5Department of Biomedical Sciences and Veterinary Public Health, Swedish University of Agricultural Sciences, 750 07 Uppsala, Sweden

**Keywords:** antibiotic, antibiotic resistance, livestock, antibiotic use, AMR, MDR, environment

## Abstract

Monitoring antimicrobial resistance (AMR) and use (AMU) is important for control. We used *Escherichia coli* from healthy young calves as an indicator to evaluate whether AMR patterns differ between Swedish organic and conventional dairy herds and whether the patterns could be related to AMU data. Samples were taken twice, in 30 organic and 30 conventional dairy herds. Selective culturing for *Escherichia coli*, without antibiotics and with nalidixic acid or tetracycline, was used to estimate the proportions of resistant isolates. Microdilution was used to determine the minimum inhibitory concentrations (MICs) for thirteen antimicrobial substances. AMU data were based on collection of empty drug packages. Less than 8% of the bacterial growth on non-selective plates was also found on selective plates with tetracycline, and 1% on plates with nalidixic acid. Despite some MIC variations, resistance patterns were largely similar in both periods, and between organic and conventional herds. For most substances, only a few isolates were classified as resistant. The most common resistances were against ampicillin, streptomycin, sulfamethoxazole, and tetracycline. No clear association with AMU could be found. The lack of difference between organic and conventional herds is likely due to a generally good animal health status and consequent low AMU in both categories.

## 1. Introduction

Antimicrobial resistance (AMR) in bacteria is a natural phenomenon that is accelerated by the selection pressure caused by antimicrobial use (AMU). Antibacterial drugs are important tools in human and veterinary medicine, necessary to combat bacterial infections, conduct advanced surgical and immunosuppressive treatments as well as ensure global food security [1]. Overuse and misuse of antibacterial drugs in humans, companion animals and livestock promote antimicrobial resistance worldwide, leading to an increased risk of treatment failures [1]. Both veterinarians and physicians face the challenge of balancing the need to treat infections against the risk of promoting AMR. Historically, Sweden has been a strong advocate for developing and implementing strategies to reduce the selection pressure by reducing AMU and closely monitoring AMU and AMR across all sectors [2,3,4]. Monitoring AMU in animals is challenging in many ways, as regards legislative framework, data sources and methods for data collection and analysis [5,6,7]. The quality and availability of data on livestock AMU vary between different regions of the world but many countries have spent a considerable effort on the development of data collection systems [5]. In Sweden, as in the rest of the European Union, antibiotics for animals are available on veterinary prescription only. Swedish AMU statistics stem from sales data from pharmacies, based on veterinary prescriptions [8]. Generally, farm-based data are not available in national statistics. In the Swedish dairy sector, however, detailed statistics on animal health and veterinary treatments are available and continuously evaluated.

The main reason for studying AMU is to monitor strategies to contain AMR, and to assess associations with AMR prevalence. Harada and Asai [9] reviewed data on AMR prevalence in bacteria from cattle in several countries around the world. Out of the included countries, Sweden presented the lowest prevalence figures, whereas countries such as Japan, France and Germany had medium levels and the Netherlands had the highest prevalence. AMR monitoring entails many methodological challenges. In the EU, data on AMR in zoonotic and indicator bacteria from humans, selected animal species and food are collected annually and jointly analyzed [6,7]. The AMR figures are based on epidemiological cut offs, so-called ECOFFs [10]. According to Commission Decision 213/652/EU, antimicrobial susceptibility testing of *Escherichia coli* from fecal samples taken at slaughter from calves <1 year of age should be included in AMR monitoring. This, however, applies only to countries where the total amount of meat from calf slaughter exceeds 10,000 tones/year and there is not sufficient longitudinal data to evaluate temporal trends in those countries that participate in harmonized monitoring [7]. In Sweden, indicator bacteria such as *E. coli* are isolated from the intestinal content of healthy pigs and poultry sampled at slaughter within the monitoring framework, or from feces collected from live animals in other projects [8]. In 2017, *E. coli* isolated from rectal swabs taken in a project, from 85 calves <2 months old, were included in the monitoring report [11]. Approximately half of the isolates in this study were susceptible to all substances tested while less than one-third were resistant to three or more substances. The most common resistance traits were against streptomycin, sulfamethoxazole, tetracycline or ampicillin. No isolate was resistant to cefotaxime, ceftazidime, colistin, florfenicol or gentamicin. Currently, these are the only available data on indicator *E. coli* from cattle in the national monitoring program.

There are no published papers that compare AMR prevalence in different dairy production systems in Sweden. Due to the link between AMU and AMR, it would be assumed that herd types with lower AMU would also have a lower prevalence of AMR. There are some international studies comparing AMU and AMR in organic and conventional herds. In the United States, organic farms are not allowed to treat with antibiotics at all, as was confirmed in a study on AMU [12,13]. In the European regulation for organic dairy herds (Council Regulation (EC) No 834/2007), AMU is restricted to a maximum number of three treatments per cow and year. The rules of the Swedish organic certification association are in some respects stricter than the EU regulations, such as requiring double withdrawal periods for milk from treated cows [14]. Whether these strict regulations are reflected in the prevalence of AMR in Swedish organic dairy herds is not known.

The aim of this study was to estimate the prevalence of AMR in Swedish dairy herds, using susceptibility testing of *E. coli* from healthy young calves as an indicator to evaluate whether AMR patterns differ between organic and conventional dairy herds and whether they could be related to AMU data.

## 2. Materials and Methods

### 2.1. Study Population

A convenience sampling design was used, where 30 organic and 30 conventional dairy herds were selected. The herds were located throughout Sweden, with an equal number of organic and conventional farms in each geographic area, and with herd sizes reflecting the overall population of Swedish dairy herds. Each farm was visited by the first author (KS) during the indoor season, February to May in 2016 and November 2016 to March in 2017.

### 2.2. Faecal Sampling

At the beginning of each study period, fecal samples from healthy calves (i.e., calves with no signs of disease) were collected by the first author. The aim was to sample 5 calves less than two months old, but if this was not possible at the time of the visit, the farmer took the remaining samples according to instructions provided and sent the samples to the laboratory. The fecal samples were collected from rectum with Amie’s charcoal culture swabs (Copan diagnostics Inc., Murrieta, CA, USA) and either brought directly to the laboratory by the first author, or sent by standard mail. The samples were, upon arrival to the lab, stored in a refrigerator and analyzed within 48 h from sampling.

At the beginning of the second study period, the first author also collected fecal samples from the farm environment. Two fecal swab samples were collected: one from an indoor drainage site and one from the manure pit. E-swabs (Copan diagnostics Inc., Murrieta, CA, USA) were used for sampling, and the samples were brought to the laboratory by the first author and stored as described above.

### 2.3. Collection of AMU Data

In a parallel study, information about AMU was collected from the study herds during three months following each initial sampling visit [15]. On-farm data collection was performed according to the so-called BIN method, where empty drug containers were collected on each farm as described by Olmos Antillón et al. [15]. Briefly, the farm staff/owners were instructed to place discarded packaging of any drug used on farm (administered by them or a visiting veterinarian) into plastic bags throughout the observation period. The bags were collected one day after the end of each observation period. In the current study, these data were used as a proxy for AMU in the study herds.

### 2.4. Antimicrobial Susceptibility Testing

We used *E. coli* as the AMR indicator bacteria and the quality accredited laboratory methods described in detail by Duse et al. [16]. In summary, the samples were diluted in 3 mL of 0.9% NaCl and, subsequently, 50 µL of 10-fold dilutions (10^−2^ and 10^−4^ for calf samples, 10^−1^ and 10^−3^ for environmental samples) were streaked on Petrifilm^TM^ (3M^TM^, St Paul, MN, USA) Select *E. coli* count (SEC) plates (3M Microbiology Products) and cultured overnight at 42 °C. The colony-forming units (CFUs) were counted the next day and calculated back to CFU/mL in the original sample.

The proportion of *E. coli* that were resistant to nalidixic acid and tetracycline was determined by parallel plating of the diluted fecal samples on SEC plates supplemented with 50 µL of 672 µg/mL nalidixic acid or 1344 µg/mL tetracycline, respectively. The plates were incubated overnight at 42 °C and the CFUs were counted on the next day. Isolates growing on these plates were regarded as resistant. Based on the estimated CFU/mL in the original sample, it was possible to estimate the proportion of tetracycline-resistant and nalidixic acid-resistant *E. coli*.

Subsequently, one random colony from each sample on the plates without antibiotics was selected, subcultured and identified as *E. coli* by morphology and the indole test. The antimicrobial susceptibility of these *E. coli* isolates was then determined with a VetMIC^TM^ (SVA, Uppsala, Sweden) panel of 13 antimicrobial substances. Epidemiological cut-off values for the minimum inhibitory concentration (MIC), determined according to the European Committee on Antimicrobial Susceptibility Testing [10], were used to classify isolates as susceptible or resistant. An isolate was defined as resistant if the MIC value was above the cut off for an antimicrobial substance and as multidrug resistant (MDR) if the MIC values were above the cut off for at least three substances from different antimicrobial classes.

All isolates with an MIC for colistin >2 mg/L were examined with PCR targeting *mcr* 1–5, according to Rebelo et al. [17]. Similarly, isolates that were resistant to third-generation cephalosporines (MIC for ceftazidime >0.5 mg/L), were examined further for extended spectrum beta-lactamase (ESBL) production and ESBL genes. Phenotypic confirmatory tests for production of ESBL in *E. coli* were performed with and without clavulanic acid in Sensititre EUVSEC2 (Thermo Fisher Scientific Inc., Waltham, MA, USA) microdilution panels and interpreted according to EUCAST [10]. PCR for identification of ESBL-encoding genes was performed on ESBL-producing isolates as previously described [18].

All analyses during the first period were performed by the accredited laboratory in the Swedish National Veterinary Institute (SVA), while analyses during period 2 were performed at the Zoonosis Science Center, Uppsala University. To ensure equal procedures in both periods, the researchers performing the analyses in the second period carried out the laboratory work and interpretations of the VetMIC plates on some samples together with SVA staff. These samples included isolates where the interpretation of MIC values was difficult as well as isolates where the results were more evident.

### 2.5. Antimicrobial Resistance Patterns and Herd Production System

The difference in the proportion of resistant isolates (for each antimicrobial substance as well as MDR) between organic and conventional herds was tested with Fisher’s exact test. The difference in the proportion of *E. coli* growing on media supplemented with tetracycline or nalidixic acid, as compared to non-supplemented medium, between organic and conventional herds was assessed by the Mann Whitney U test for unpaired samples.

The mean and median MIC values and proportion of isolates classified as resistant for each substance, from herds treated or not treated with the same class of drug, were also assessed. In addition, the proportion of resistant isolates for each substance and the herd AMU for the corresponding substance, expressed as the total number of defined course doses (DCD/animal/year, see Olmos Antillón et al., 2020 for details on DCD calculations), was assessed for organic and conventional herds.

Heatmaps illustrating AMR patterns in calf isolates from each herd were generated in Python (www.python.org) using the matplotlib, pandas and seaborn packages. All Python scripts are available on request.

### 2.6. Ethical Statement

All animals in this study were treated according to the ethical standards of the Swedish regulations. The competent authorities stated that no ethical permission was required for this sampling. Participation in this study was voluntary, and the farmers were informed about the purpose and methods of this study. They were assured that all information would be treated anonymously and that they could withdraw from this study at any time.

## 3. Results

In total, 293 calves from 60 herds were sampled during the first period. During the second period, 258 calves from 54 herds were sampled (3 organic and 3 conventional farms did not participate in the second round). A total of 103 fecal environmental samples (54 from manure drainage and 49 from manure pit) were also taken from these 54 herds at the start of the second period. The herd size ranged from 48 to 230 cows, with a median of 80 cows in the organic herds and 110 in the conventional herds. The number of sampled calves per herd and time point varied from 2 to 6, with a median of 5. The age of the sampled calves ranged from 0 to 46 days, with a median of 15 days in both herd types in the first sampling round, while the median age was 18 days in organic herds and 26 days in conventional herds in the second sampling round.

### 3.1. Proportion of Tetracycline- and Nalidixic Acid-Resistant E. Coli

In the calf samples from organic herds in the first period, 7.3% of the bacterial growth on non-selective plates was also found on selective plates with tetracycline; the corresponding proportion for selective plates with nalidixic acid was 0.8%. In samples from conventional herds, 4.9% of the growth on non-selective plates was also found on selective plates with tetracycline and 1.0% on selective plates with nalidixic acid. The difference was significant (*p* < 0.05) for nalidixic acid but not for tetracycline. The results from the calf samples for period 2 had to be discarded due to methodological errors that could not be resolved. In the environmental samples from organic herds, 6.4% of the bacterial growth on non-selective plates was also found on selective plates with tetracycline, the corresponding proportion for selective plates with nalidixic acid was 2.6%. In samples from conventional herds, 9.5% of the growth on non-selective plates was also found on selective plates with tetracycline and 3.8% on selective plates with nalidixic acid. The differences observed were not significant.

### 3.2. Antimicrobial Susceptibility

Figure 1, Figure 2 and Figure 3 show the distribution of MIC values for substances in the VetMIC^TM^ panel for calf isolates in period 1 and 2, and for environmental isolates, respectively. Some variations were seen but the resistance patterns were largely similar in period 1 and 2, and between organic and conventional herds. For most of the antimicrobial agents, only a few isolates had MIC values above the epidemiological cut offs. All isolates from calves and environment were susceptible to florfenicol. All but two isolates in period 2 (one from a calf and one from a manure drainage) were susceptible to gentamicin, with the resistant isolates having MICs just above the cut off. The proportions of isolates with MICs above the epidemiological cut offs were highest for ampicillin, streptomycin, tetracycline and sulfamethoxazole (Figure 1, Figure 2 and Figure 3).

The proportions of isolates with AMR and MDR in the different samples are illustrated in Figure 4. There were no clear differences between calf samples in period 1 and period 2, whereas the proportions of both AMR and MDR were lower in the environmental samples. In period 1, 52% of the calf isolates were resistant to at least one of the tested antimicrobial substances, the corresponding figure for period 2 was 44%, and for environmental samples 27%. There was no significant difference between organic and conventional herds. In both periods, 28% of the calf isolates were MDR and in environmental samples 12% of the isolates were MDR (7% of isolates from manure drainage and 16% of isolates from manure pit). There was no significant difference between organic and conventional herds. There were very few significant differences between the herd types for single antimicrobial substances, and no consistent pattern over the two sampling occasions (Figure 1, Figure 2 and Figure 3). The most common resistances were against ampicillin, streptomycin, sulfamethoxazole, and tetracycline. Most herds had a rather high proportion of isolates that were resistant to at least one antimicrobial, but the majority had no, or very few, samples that were MDR (Figure 4). In the first period, five herds had >60% of calf isolates with MDR, while three other herds had >60% of calf isolates with MDR in the second period.

In period 1, 31 *E. coli* isolates had a colistin MIC of >2 mg/L, but no *mcr* genes were detected. Three isolates were resistant to third-generation cephalosporins, where one was confirmed as ESBL producing and carried CTX-M-1.

In period 2, 11 *E. coli* isolates from calf samples had a colistin MIC above cut off, but at the time of PCR testing, pure cultures could not be obtained from four of these. The remaining seven were subjected to the *mcr*-PCR and all were negative. Five calf *E. coli* isolates were resistant to third-generation cephalosporins, but none of these were ESBL producing and were therefore not further analyzed for resistance genes. Four isolates from environmental samples had a colistin MIC above cut off but three could not be pure cultured for the PCR, the remaining one was negative in the *mcr*-PCR. When the microdilution tests were performed, all cultures were checked for purity and hence it was concluded that the contamination had occurred after this step, in the process of storing the isolates.

The resistance patterns in calf isolates from each herd and for each substance are illustrated in the heatmaps in Figure 5. No consistent pattern could be discerned between herd type or sampling period and the heatmaps confirm the generally low resistance prevalence in the sampled herds.

### 3.3. Association between AMU and AMR

All herds but one had used some antibiotic treatment during the observation periods. Table 1 shows the average and median MICs as well as the proportions of resistant isolates of each sample type, in herds with or without records of having used the corresponding antimicrobial substance. For most substances, the average MIC and the proportion of resistant isolates was lower in herds with low or no recorded treatments, while for others the opposite, or no difference, was seen. There was little or no difference in median MIC values, and the observed differences correspond to one dilution step.

The overall proportions of resistant *E. coli* isolates from calf and environmental samples and the corresponding use of antimicrobial treatments in these herds, expressed as the average of the total number of defined course doses per animal per year for the corresponding antimicrobial substance, are shown in Table 2. Some variation but no clear association between the proportion of resistant isolates and recorded AMU or herd management category could be found.

From the available data, there was no indication of higher AMU in herds with higher proportions of MDR isolates.

## 4. Discussion

To the best of our knowledge, potential differences in AMR patterns between organic and conventional dairy herds in Sweden have not been previously studied. Some studies indicate that AMU is lower in organic herds [19,20], and AMR has been reported to be lower in organic pig herds, although the difference between organic and conventional production was less in Sweden than in some other countries [21]. Previous studies indicate that the differences in health status (that would affect AMU) between organic and conventional dairy herds in Sweden are small [22,23]. This is supported by the data from the herds in the present study that revealed no statistical difference in overall AMU between the two production systems, although some minor difference in patterns of AMU could be noted [15].

We chose to sample calves because previous data indicate that the prevalence of AMR decreases with the age of the animals [13,24,25] and we wanted to maximize the detection of AMR in the study herds. Sampling older animals might underestimate the levels of AMR, which should be taken into account when comparing data from different studies [24]. The environmental manure samples collected at the start of period 2 can be assumed to represent older animals (as cows produce most of the manure in the herd) and reflect a previous time period. The lower proportion of isolates classified as resistant in these samples, in comparison to the calf samples, support the assumption that herd-level AMR might be underestimated in samples from older animals. On the other hand, the storage time for the manure in the farm environment may also affect the composition and resistance pattern of the *E. coli* population in the samples.

In Europe, the prevalence of ESBL-producing *E. coli* in food-producing animals varies by country and animal species. In 2017, the prevalence in individual veal calves ranged from 7.1% in Denmark to 89.0% in Italy, with an EU mean of 44.5% [6]. Overall, in the eight EU countries supplying data on resistance in *E. coli* from calves to ampicillin, cefotaxime, ciprofloxacin and tetracycline for the period 2009–2017, there were 10 decreasing and 8 increasing trends [7]. A study in Spanish cattle herds tested *E. coli* isolated from healthy animals of all ages and found a significantly higher prevalence of resistance in dairy herds than in beef herds [26]. As much as 97.8% of all isolates were resistant to fourth-generation cephalosporins, 87.4% were resistant to third-generation cephalosporins, 70.4% to tetracycline, 70.4% to sulfamethoxazole, 47.4% to trimethoprim, 41.5% to ciprofloxacin, 28.9% to chloramphenicol and 23.7% to gentamicin.

In a Chilean study, *E. coli* isolated from healthy dairy calves, calves with diarrhoea and their environment (bedding) showed 92% resistance to amoxicillin, 18.3% to ceftiofur, 27.5% to enrofloxacin, 25.5% to florfenicol, 7.2% to gentamicin, 53.6% to oxytetracycline and 37.5% to trim-sulfa [27]. Nearly half of the isolates (49%) were resistant to three or more substances. In a case-control study from 2012 in healthy and diarrheic calves in Swedish dairy herds, the corresponding figures were 25% resistant to ampicillin, 0% to ceftiofur, 14% to enrofloxacin, 0% to florfenicol, 0% to gentamicin and 32% to tetracycline, with 28% resistant to 3 or more substances [28].

From an international perspective, AMR levels are low in Sweden [8] and this was also confirmed in our study, with AMR figures well below the levels found in the studies cited above. As seen in Figure 5, the overall pattern in the sampled herds reflect a low level of AMR. The highest number of isolates with MIC values above epidemiological cut offs were seen for ampicillin, streptomycin, sulfamethoxazole and tetracycline. However, only a small percentage of colonies grew on the plates supplemented by tetracycline at the cut-off concentration, indicating that the selected isolates with higher MIC values constituted a minority of the *E. coli* present in the samples.

The most common cause for AMU in Swedish dairy herds is mastitis [29]. In 2018/2019, the reported treatment incidence for clinical mastitis in dairy cows was 9.0 recorded treatments per 100 lactations [29]. Benzylpenicillin is the most common drug used and is recorded for more than 90% of reported treatments in the period 2018–2019 [29]. Historically, tetracycline was one of the most commonly used antimicrobial substances in Swedish livestock. In the last decade, the overall sale of tetracycline for veterinary use in Sweden has halved [8]. Still, tetracycline, together with sulfonamides and trimethoprim are among the most frequently used substances in dairy herds, although with treatment incidences at less than 10% of the beta lactam use [29]. Nalidixic acid is a representative for quinolones, enrofloxacin is the only quinolone used in Swedish animals. Enrofloxacin is only recommended for treatment of mastitis if *Klebisella* spp. is confirmed, while for *E. coli* mastitis only supportive therapy is recommended [30]. Since 2012, legislation restricts the use of quinolones and third- and fourth-generation cephalosporins in animals to situations when bacterial culture and susceptibility testing demonstrate that there is no other effective treatment option [30]. The level of quinolone-resistant isolates was low in this study, and the results from selective culturing indicate that the resistant isolates constituted a minority of the *E. coli* in the samples. No treatment of dairy cattle with cephalosporins was reported in the last available statistics from the Swedish dairy association [29]. Only one ESBL-producing isolate out of eight cephalosporin-resistant *E. coli* was detected in this study, which supports the absence of a selective pressure (i.e., exposure to third- or fourth-generation cephalosporins) in the studied herds. Chloramphenicol is not allowed in food-producing animals in Sweden, but florfenicol is registered for use in cattle [31]. Veterinary treatment guidelines, however, do not recommend its use [29], and no treatment of dairy cattle with florfenicol is reported in the available statistics [29]. A total of 28 isolates had MICs above cut off for chloramphenicol. Chloramphenicol can be produced by *Streptomyces venezuelae* in soil and absorbed by grass and crops. If animals are fed roughage and crops/grains that have grown in soil where *Str. venezuelae* is present, it may be a source for chloramphenicol exposure of these animals [32]. Another reason for chloramphenicol resistance in the absence of a selective pressure could be the location of chloramphenicol resistance genes on transferable genetic elements that carry other resistance genes, causing co-resistance to other drugs where a selective pressure may exist. Co-resistance to chloramphenicol and other drug classes commonly used in cattle, such as dihydrostreptomycin and trimethoprim, was reported in bovine *E. coli* in a Japanese study [33] and co-resistance to tetracycline was found in >92% of chloramphenicol resistant *E. coli* from humans and animals in a US survey [34].

A total of 46 isolates were defined as colistin resistant according to MIC values but no *mcr* gene was detected. Colistin is not used in Swedish cattle [29] and there is no colistin-containing drug preparation registered for use in cattle in Sweden [31]. So far, *mcr* genes constitute the only known transferable genetic element conferring colistin resistance, so the high MIC values may be caused by chromosomal mutations or a yet unidentified mobile genetic element [35], or may be due to the inherent difficulties in MIC analysis for colistin. The microdilution method for colistin MIC determination is challenging, as the concentration of free colistin may be affected by adherence to organic or inorganic material, or the presence of polysorbate in the dilution broth [36]. Within the scope of this study, we cannot determine whether some of the high MIC values were due to a lower than expected concentration of colistin in the MIC plates or whether the isolates were indeed colistin resistant.

The differences in average and median MIC values (see Table 1) are not surprising as the susceptibility testing method uses a series of 2-fold dilutions of the antimicrobial substance and hence a difference in one dilution step will cause a 2-fold difference in MIC. Repeated testing of the same isolate may result in a one-step change in MIC [10] and this should be kept in mind when interpreting results that are close to the cut-off value, regardless of whether ECOFFs or clinical breakpoints are used. AMR levels can be presented in various ways, but the most transparent is showing the MIC distributions, as in Figure 1, Figure 2 and Figure 3. By showing the distribution of the obtained MIC values when testing a number of non-clinical isolates collected in a systematic manner, it might be possible to discern two phenotypic populations with varying levels of MIC. The epidemiological cut-off points provided by EUCAST are based on a large number of samples and aim to differentiate between the wild type of a bacterial population and isolates with acquired resistance, but there may be overlap in the MIC distributions of these two types of isolates [10]. Hence, cut offs or breakpoints may be regarded as guidance, not an absolute divider between wild-type isolates and isolates with acquired resistance (potentially due to exposure to antimicrobials). The representativeness of the isolates is also an issue, although single isolates from a few animals per herd are regularly used as a basis for illustrating the herd-level AMR pattern, or for national monitoring [7]. The results from our selective culturing, where the resistant isolates constituted a minority of the *E. coli* population in most of the samples, illustrate the challenge of obtaining representative isolates. Methodological variation caused by differences in sample transportation time or different people taking the samples was not expected to have affected the results, and no systematic differences related to these aspects were observed.

AMU contributes to AMR by exerting a selective pressure on bacterial populations, favoring clones carrying resistance traits that protect them from the antimicrobial substances used. The varying patterns of phenotypic resistance traits and sometimes contradictory relationship between AMR and AMU for specific substances seen in this study illustrate the complexity of AMR dynamics. The optimal timeframe for assessing the selective effect of AMU is difficult to pinpoint. We chose to use on farm-collected AMU information for two time periods during the indoor season in order to reflect the overall pattern of AMU in the study herds. The AMU data did not cover the entire time period before sampling but were deemed to be the most accurate reflection of actual on-farm use, not hampered by challenges in reporting [15]. One of the study farms stated to not have used any antibiotics for at least five, maybe as much as 10 years, either in animals (pets included) or people living on the farm. Still, resistant isolates of *E. coli* were found in the samples from this farm, demonstrating the unpredictability of temporal AMR patterns. The association between AMU and AMR may be apparent on a larger scale, although not easily demonstrated on an individual farm level. A recent European study in poultry, pigs and veal calves detected significant associations between on-farm AMR and national AMU for some substances in some herd types but not all [37].

AMR has been detected in bacteria isolated from flies, rats and other animals in farm environments [38], further illustrating the challenges in determining the associations between AMU and AMR. Similar patterns of ESBL-producing *E. coli* have been demonstrated in wild gulls and humans, and it has been suggested that the humans might have served as the source for AMR in animals [39]. A possible route for cattle is from people excreting antimicrobial residues (and resistant bacteria) into effluent [40] and further into surface water that is subsequently consumed by grazing livestock.

In the present study, there was no obvious difference in AMR patterns in organic and conventional herds, indicating that a generally good animal health status and consequent low AMU may have the same effect in controlling AMR as the specific AMU rules for organic production. Although the number of farms and sampling occasions were limited and small differences may have been difficult to discern, the strict regulation of AMU in both production systems in Sweden prompts the question whether other herd-level factors exert a higher influence on AMR patterns in this context. Further studies of the entire farm environment are needed to disentangle the complex web of AMR and its drivers on livestock farms.

## 5. Conclusions

No obvious difference in AMR patterns between organic and conventional herds could be detected in this study, most likely due to a generally good animal health status and consequent low AMU in both herd types. The results illustrate the general variation in AMR patterns on a farm level and the challenges in detecting associations with herd management or AMU. Further studies of the entire farm environment are needed to disentangle the complex web of AMR and its drivers on livestock farms.

## Figures and Tables

**Figure 1 antibiotics-09-00834-f001:**
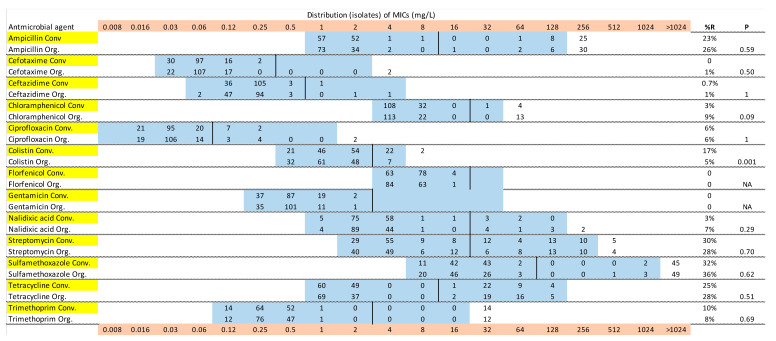
Minimum inhibitory concentration (MIC) distribution of *Escherichia coli* isolates from calves, sampled in period 1, in 30 (148 isolates) organic (Org) and 30 (145 isolates) conventional (Conv) Swedish dairy herds, in a panel of 13 antimicrobial agents. Blue color indicates the range of concentrations tested. Vertical black lines are the epidemiological cut-off points, according to EUCAST. *p* values test the difference between organic and conventional herds with Fisher’s exact test. Yellow color denotes isolates from conventional herds and pink color shows MIC values.

**Figure 2 antibiotics-09-00834-f002:**
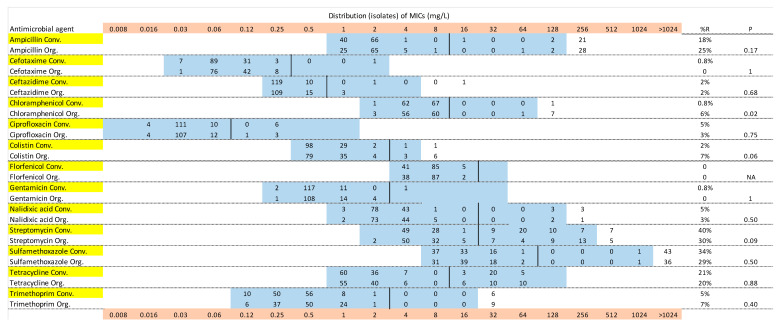
Minimum inhibitory concentration (MIC) distributions of *Escherichia coli* isolates from calves, sampled in period 2, in 27 (127 isolates) organic (Org) and 27 (131 isolates) conventional (Conv) Swedish dairy herds, in a panel of 13 antimicrobial agents. Blue color indicates the range of concentrations tested. Vertical black lines are the epidemiologic cut-off points according to EUCAST. *p* values test the difference between organic and conventional herds with Fisher’s exact test. Yellow color denotes isolates from conventional herds and pink color shows MIC values.

**Figure 3 antibiotics-09-00834-f003:**
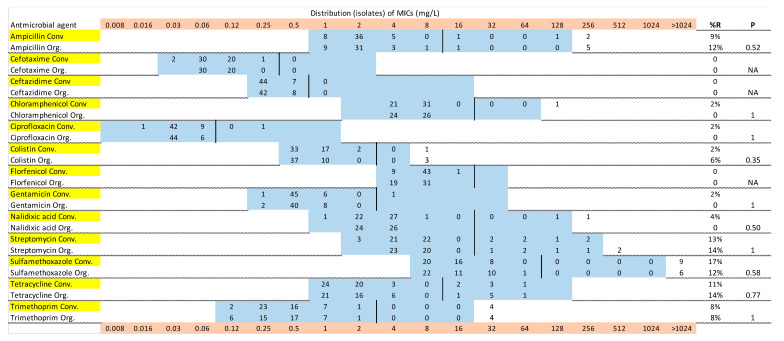
Minimum inhibitory concentration (MIC) distributions of *Escherichia coli* isolates from environmental samples of farm manure (drainage and manure pit) at the beginning of period 2, in 27 (50 isolates) organic (Org) and 27 (53 isolates) conventional (Conv) Swedish dairy herds, in a panel of 13 antimicrobial agents. Blue color indicates the range of concentrations tested. Vertical thicker black lines are the epidemiologic cut-off points according to EUCAST. *p* values test the difference between organic and conventional herds with Fisher’s exact test. Yellow color denotes isolates from conventional herds and pink color shows MIC values.

**Figure 4 antibiotics-09-00834-f004:**
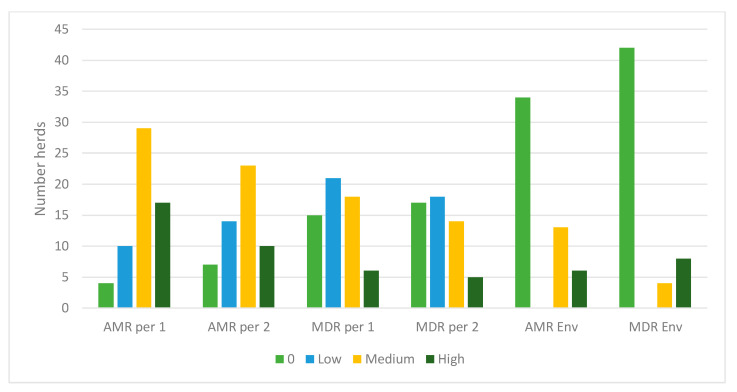
Number of sampled herds with zero (0), low (1–25%), medium (26–60%) and high (>60%) proportions of *Escherichia coli* with antimicrobial resistance to any antimicrobial class (AMR) or to three or more antimicrobial classes (MDR) in Swedish organic and conventional dairy herds. Isolates from calf samples in period 1 (per 1) and period 2 (per 2) and environmental samples in period 2 (env).

**Figure 5 antibiotics-09-00834-f005:**
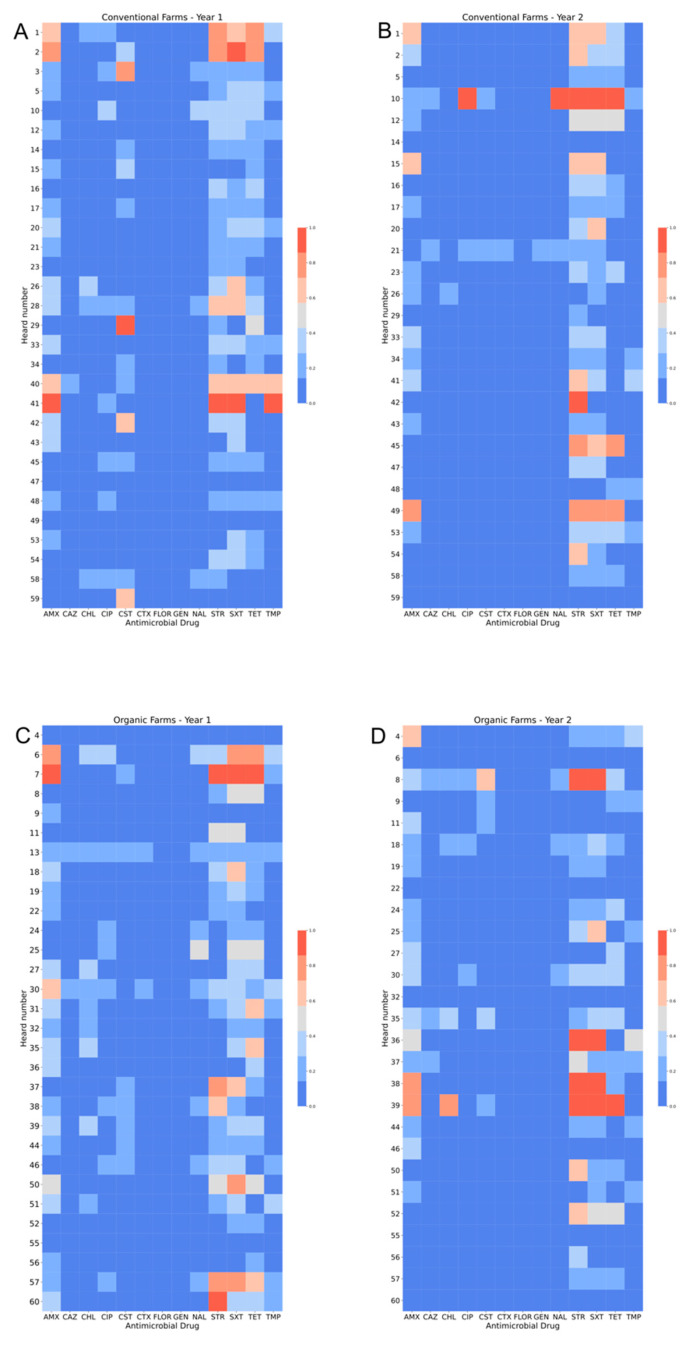
Heatmaps showing the patterns of resistant and susceptible *Escherichia coli* isolates from Swedish organic and conventional dairy herds sampled in two different time periods. (**A**) conventional herds 1st sampling (**B**) conventional herds 2nd sampling (**C**) organic herds 1st sampling (**D**) organic herds 2nd sampling. Color scale illustrates average value for each herd and each antimicrobial substance tested, where 1 = resistant and 0 = susceptible. AMX = ampicillin, CAZ = ceftazidime, CHL = chloramphenicol, CIP = ciprofloxacin, CST = colistin, CTX = cefotaxime, FLOR = florfenicol, GEN = gentamycin, NAL = nalidixic acid, STR = streptomycin, SXT = sulfamethoxazole, TET = tetracycline, and TMP = trimethoprim.

**Table 1 antibiotics-09-00834-t001:** Summary of antimicrobial susceptibility test results for *Escherichia coli* isolates from fecal samples from calves and environment (manure drainage and manure pit) in 27 organic and 27 conventional Swedish dairy herds (sampling period 2) and the corresponding use of antimicrobial treatments in these herds, as determined by collection of empty drug packages. Treated=use of the corresponding substance recorded for the herd. Not treated=no use of the corresponding substance recorded.

Substance	Calf Isolates		Environmental Isolates	
	Herd Treated ^a^	Herd Not Treated	Herd Treated	Herd Not Treated
**Ampicillin ^a1^**	N = 53	N = 1	N = 53	N = 1
Average MIC	54	2	22	3
Median MIC	2	2	2	2
%resistant	22	0	12	0
**Ciprofloxacin ^a2^**	N = 6	N = 48	N = 6	N = 48
Average MIC	0.055 ^b^	0.036	0.040	0.035
Median MIC	0.030	0.030	0.030	0.030
%resistant	14	2	7	2
**Nalidixic acid ^a2^**	N = 6	N = 48	N = 6	N = 48
Average MIC	21	7	3	6
Median MIC	4	2	5	4
%resistant	10	2	0	2
**Gentamicin ^a3^**	N = 20	N = 34	N = 20	N = 34
Average MIC	0.60	0.56	0.58	0.71
Median MIC	0.50	0.50	0.50	0.50
%resistant	0.5	0	0	4
**Streptomycin ^a3^**	N = 20	N = 34	N = 20	N = 34
Average MIC	69	67	42	10
Median MIC	8	8	8	6
%resistant	32	44	20	10
**Tetracycline ^a4^**	N = 10	N = 44	N = 10	N = 44
Average MIC	7	10	5	3
Median MIC	2	2	2	2
%resistant	19	23	16	13
**Sulfamethoxazole ^a5^**	N = 8	N = 46	N = 8	N = 46
Average MIC	850	590	463	264
Median MIC	320	160	80	160
%resistant	42	29	22	14
**Trimethoprim ^a5^**	N = 8	N = 46	N = 8	N = 46
Average MIC	2.22	2.31	5.86	2.32
Median MIC	0.50	0.50	0.50	0.50
%resistant	6	6	17	8

^a^ Treatments represent the following corresponding substances: ^a1^ benzylpenicillin/amoxicillin/kloxacillinbensatin; ^a2^ enrofloxacin; ^a3^ dihydrostreptomycin; ^a4^ oxytetracycline; ^a5^ sulfadiazin-trimethoprim/sulfadoxine-trimethoprim, N = number of herds with recorded treatment of the corresponding substance, and ^b^ number of decimal points reflect the number of decimal points in the MIC values (i.e., depending on the concentration/degree of dilution in the test panel).

**Table 2 antibiotics-09-00834-t002:** Proportion of resistant *Escherichia coli* isolates from fecal samples from calves and environment (manure drainage and manure pit) in 27 organic and 27 conventional Swedish dairy herds (sampling period 2) and the corresponding use of antimicrobial treatments in these herds, as determined by collection of empty drug packages. Organic = organic herds, conventional = conventional herds, %R = proportion of resistant isolates (%), and DCD = total number of defined course doses per animal per year for the corresponding antimicrobial substance, expressed as the average figure for all herds in the category during the two data collection periods.

Substance	Calf Isolates		Environmental Isolates	
	Organic	Conventional	Organic	Conventional
Ampicillin %R	25	18	12	9
DCD penicillins	0.584	1.170	0.584	1.170
Ciprofloxacin %R	3	5	0	2
Nalidixic acid %R	3	5	0	4
DCD enrofloxacin	0.002	0.002	0.002	0.002
Gentamicin %R	0	0.8	0	2
Streptomycin %R	30	40	14	13
DCD dihydrostreptomycin	0.102	0.167	0.102	0.167
Tetracycline %R	20	21	14	11
DCD oxitetracycline	0.010	0.005	0.010	0.005
Sulfamethoxazole %R	29	34	12	17
Trimethoprim %R	7	5	8	8
DCD trimethoprim/sulfa	0.021	0.012	0.021	0.012

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
