# Peer review of "Antimicrobial Resistance Patterns in Organic and Conventional Dairy Herds in Sweden"

_antibiotics, 2020, doi:10.3390/antibiotics9110834_

Round 1

Reviewer 1 Report

The manuscript estimates the prevalence of AMR in Swedish dairy herds, using susceptibility testing of E. coli from healthy young calves as an indicator, to evaluate if the AMR  patterns differ between organic and conventional dairy herds and if they could be related to AMU data.

Although single E. coli isolates from a few animals per herd are used as a basis for illustrating the herd-level AMR pattern, or for national monitoring, more useful data would be obtained from investigating AMR in clinical isolates. The most common cause for AMU in the world, and also in Swedish dairy herds, is mastitis. Therefore, a more interesting study would be to compare AMR prevalence of the principal pathogens implicated in mastitis (Staphylococcus aureus, coagulase-negative staphylococci (CNS) and Streptococcus species) in different dairy production systems, organic and conventional herds, instead of AMR prevalence of non-clinical isolates.

Other considerations:

Figure 4. Perhaps, it may not be necessary.

Table 1. The information is not completely clear.

Discussion

It would be convenient to cite and discuss, perhaps, some previous related works such as for example:

de Verdier K, Nyman A, Greko C, Bengtsson B. Antimicrobial resistance and virulence factors in Escherichia coli from Swedish dairy calves. 2012. Acta Vet Scand. Jan 26;54(1):2. doi: 10.1186/1751-0147-54-2.

Ceccarelli D, Hesp A, van der Goot J, Joosten P, Sarrazin S, Wagenaar JA, Dewulf J, Mevius DJ, Effort Consortium OBOT. Antimicrobial resistance prevalence in commensal Escherichia coli from broilers, fattening turkeys, fattening pigs and veal calves in European countries and association with antimicrobial usage at country level. J Med Microbiol. 2020 Apr;69(4):537-547. doi: 10.1099/jmm.0.001176.

Author Response

Although single E. coli isolates from a few animals per herd are used as a basis for illustrating the herd-level AMR pattern, or for national monitoring, more useful data would be obtained from investigating AMR in clinical isolates. The most common cause for AMU in the world, and also in Swedish dairy herds, is mastitis. Therefore, a more interesting study would be to compare AMR prevalence of the principal pathogens implicated in mastitis (Staphylococcus aureus, coagulase-negative staphylococci (CNS) and Streptococcus species) in different dairy production systems, organic and conventional herds, instead of AMR prevalence of non-clinical isolates.

Authors’ response: Although we understand the point made by the reviewer, we believe that comparing clinical isolates would be a very different approach that might be more suitable for a study focusing on mastitis and health problems in different production systems, rather than general monitoring of AMR in different production systems. We applied standard monitoring methods using commensal indicator bacteria that allow for an indication of the magnitude of the selective pressure/AMR prevalence in an animal population. Specifically, ECOFFs classify isolates with acquired reduced susceptibility as non-wild type/resistant. This classification is relevant for monitoring purposes, whereas studying clinical isolates and using clinical breakpoints is more relevant for disease monitoring and follow-up on treatment guidelines. Hence, for the purpose of our study we believe that the methods used were the most appropriate, while another study looking at clinical isolated would also be interesting.

Figure 4. Perhaps, it may not be necessary.

Authors’ response: Although the figure is not strictly necessary, we believe it provides visual information that is not easily gained from the text or the other figures. The complexity of the AMR patterns over time in the study herds is illustrated in this figure and hence we decided to use it in the graphical abstract suggested by reviewer 3.

Table 1. The information is not completely clear.

Authors’ response: We are unsure what the problem is but have tried to simplify the treatment explanations for this table (lines 204-209), as they may have played a part. We can make further clarifications if needed.

Discussion

It would be convenient to cite and discuss, perhaps, some previous related works such as for example:

de Verdier K, Nyman A, Greko C, Bengtsson B. Antimicrobial resistance and virulence factors in Escherichia coli from Swedish dairy calves. 2012. Acta Vet Scand. Jan 26;54(1):2. doi: 10.1186/1751-0147-54-2.

Ceccarelli D, Hesp A, van der Goot J, Joosten P, Sarrazin S, Wagenaar JA, Dewulf J, Mevius DJ, Effort Consortium OBOT. Antimicrobial resistance prevalence in commensal Escherichia coli from broilers, fattening turkeys, fattening pigs and veal calves in European countries and association with antimicrobial usage at country level. J Med Microbiol. 2020 Apr;69(4):537-547. doi: 10.1099/jmm.0.001176.

Authors’ response: The suggested publications have been added to the Discussion lines 271-274, and lines 363-366 (and the reference list).

Reviewer 2 Report

In this manuscript, the authors evaluated whether AMR patterns differ between Swedish organic and conventional dairy herds and whether the patterns could be related to AMU data. Using E. coli from healthy young calves as an indicator, they found no obvious difference in the AMR patterns in organic and conventional herds beside some MIC variations and no clear association with AMU. They also suggest that this lack of difference between organic and conventional herds is likely due to good animal health and consequent low AMU in both categories. Overall, the manuscript is well written, with detailed methods and results. On the other hand, it could be improved if the authors could clarify the questions below.

  1. The authors mentioned a stricter Swedish rule than the EU regulations for organic dairy herds, but did not clarify whether the herds in current study experienced any treatments with antibiotics or not. If possible, please clarify.
  2. Though the authors took actions to minimize technical variations for analysis during the first and second periods, there are still some that may lead to variations. For example, the potential food differences taken by different herds, the sample collection sites by different persons/studies, the time length from sample collection to arrival at the lab by different methods (personal delivery or mail) etc. Please indicate potential influences of such factors to results.
  3. How do you define healthy calves in this study?
  4. Will it be a good idea to test the fecal microbiome of these calves and associate that with the data in current manuscript?

Author Response

The authors mentioned a stricter Swedish rule than the EU regulations for organic dairy herds, but did not clarify whether the herds in current study experienced any treatments with antibiotics or not. If possible, please clarify.

Authors’ response: All herds but one had used antibiotics, as seen in table 1 and on line 356. We realise that this may not be obvious to the reader and have clarified in the Results section (line 193).

Though the authors took actions to minimize technical variations for analysis during the first and second periods, there are still some that may lead to variations. For example, the potential food differences taken by different herds, the sample collection sites by different persons/studies, the time length from sample collection to arrival at the lab by different methods (personal delivery or mail) etc. Please indicate potential influences of such factors to results.

Authors’ response: We have added some comments to the discussion (lines 348-351). We don’t think these aspects would have had any major influence on the results, differences as regards feed were expected to be related to the production type (feed used in organic herds is produced similarly as regards aspects related to AMR selection) sample transportation would not be expected to systematically affect AMR prevalence in the sample and the sampling method was simple enough not to be significantly affected by who took the sample (and most samples were taken by the first author).

How do you define healthy calves in this study?

Authors’ response: healthy calves are calves without clinical signs of disease. This is clarified on lines 392-393.

Will it be a good idea to test the fecal microbiome of these calves and associate that with the data in current manuscript?

Authors’ response: That might have been a good idea, but unfortunately the original sample material is no longer available, only the isolates.

Reviewer 3 Report

This study investigated if AMR patterns differ between Swedish organic and conventional dairy herds.The theme of manuscript is scientifically sound and the title introduces readers to general context.However, minor changes should be made before manuscript could be acceptable for publication.

The authors should better rewrite the conclusions of this study at the end of the abstract.

The authors also should re-write the conclusions to make it more informative and impactful.

Additionally, could these data also be reproduced to other countries? Please clarify.

Please refer to doi:10.3390/antiox9080693.

Author Response

The authors should better rewrite the conclusions of this study at the end of the abstract.

Authors’ response: As the word limit is reached for the abstract, we cannot squeeze in more into the conclusion and we are reluctant to change it in the abstract as we don’t want to overemphasize the results or mislead the reader in this brief section. However, if this is required we will of course rephrase the conclusion and/or shorten other parts of the abstract.

The authors also should re-write the conclusions to make it more informative and impactful.

Authors’ response: We have added a sentence to the conclusion to make it more informative (line 485-486). As we don’t want to overemphasize the importance of our results, we are unsure of how to make it more impactful.

Additionally, could these data also be reproduced to other countries? Please clarify.

Authors’ response: We are not sure if we understand this comment correctly, the Discussion compares the results with what has been found in other countries. The study could easily be reproduced in another country, using our description in the M&M section, but the resulting data would most likely be different. We have not added anything in the text, as we are not sure what was intended but will of course do so if some clarification can be provided.

Please refer to doi:10.3390/antiox9080693.

Authors’ response: We assume that this refers to the graphical abstract (as the cited study concerns a different subject). We have added a graphical abstract, thank you for the suggestion. See attachment
